# Personalized Treatment of Glioblastoma: Current State and Future Perspective

**DOI:** 10.3390/biomedicines11061579

**Published:** 2023-05-30

**Authors:** Alen Rončević, Nenad Koruga, Anamarija Soldo Koruga, Robert Rončević, Tatjana Rotim, Tihana Šimundić, Domagoj Kretić, Marija Perić, Tajana Turk, Damir Štimac

**Affiliations:** 1Department of Neurosurgery, University Hospital Center Osijek, 31000 Osijek, Croatia; 2Faculty of Medicine, Josip Juraj Strossmayer University of Osijek, 31000 Osijek, Croatia; 3Department of Neurology, University Hospital Center Osijek, 31000 Osijek, Croatia; 4Department of Diagnostic and Interventional Radiology, University Hospital Center Osijek, 31000 Osijek, Croatia; 5Department of Nephrology, University Hospital Center Osijek, 31000 Osijek, Croatia; 6Department of Cytology, University Hospital Center Osijek, 31000 Osijek, Croatia; 7Department of Radiology, National Memorial Hospital Vukovar, 32000 Vukovar, Croatia

**Keywords:** glioblastoma, chemotherapy, immunotherapy, neurosurgery, radiotherapy

## Abstract

Glioblastoma (GBM) is the most aggressive glial tumor of the central nervous system. Despite intense scientific efforts, patients diagnosed with GBM and treated with the current standard of care have a median survival of only 15 months. Patients are initially treated by a neurosurgeon with the goal of maximal safe resection of the tumor. Obtaining tissue samples during surgery is indispensable for the diagnosis of GBM. Technological improvements, such as navigation systems and intraoperative monitoring, significantly advanced the possibility of safe gross tumor resection. Usually within six weeks after the surgery, concomitant radiotherapy and chemotherapy with temozolomide are initiated. However, current radiotherapy regimens are based on population-level studies and could also be improved. Implementing artificial intelligence in radiotherapy planning might be used to individualize treatment plans. Furthermore, detailed genetic and molecular markers of the tumor could provide patient-tailored immunochemotherapy. In this article, we review current standard of care and possibilities of personalizing these treatments. Additionally, we discuss novel individualized therapeutic options with encouraging results. Due to inherent heterogeneity of GBM, applying patient-tailored treatment could significantly prolong survival of these patients.

## 1. Introduction

Glioblastoma (GBM) is the most common primary malignancy of the brain in adults, accounting for almost half of all malignant central nervous system (CNS) tumors [1]. Despite major advances in understanding molecular and genetic characteristics of GBM, significant improvement in survival of patients has not yet been achieved. Untreated patients have a median survival of only three to four months [2], whereas patients who undergo current first-line treatment, which consists of surgery, concomitant radiochemotherapy, and maintenance chemotherapy present with a median survival of approximately 15 months [1]. Evidently, patients who undergo RT and chemotherapy after the surgery fare better than those who undergo surgery alone [3]. There are differing reports regarding incidence of GBM—in the adult population it is estimated to be in the range between 3.19 [4] and 4.17 [1,5] per 100,000 person-years. Similar to many other malignancies, the incidence and worse prognosis of GBM rises with increasing age [6]. Improving overall survival (OS) as well as progression-free survival (PFS) for GBM patients is still a great challenge. In this review, we will present detailed first-line treatment of GBM, as well as novel therapeutic options that are still in development with promising results. We discuss that even the current therapeutic options of GBM can be personalized. However, innovations in biomedical technology, such as patient-derived brain organoids, can lead the pursuit towards individualized GBM therapy.

## 2. Surgical Treatment

Surgery presents the first step of GBM treatment, and the role of surgery is two-fold. Primarily, surgical resection reduces the tumor volume and improves the OS [7]. Secondarily, during surgery neurosurgeons obtain tumor samples which are then used to determine the diagnosis of the neoplasm—based on the most recent classification according to the World Health Organization (WHO) [8]. It should be emphasized that obtaining adequate samples is indispensable for subsequent treatments—and for the concept of personalized treatment. Significant technological improvements have been achieved to improve surgical resection, such as the use of intraoperative neuronavigation [9] and fluorescence-guided resection [10]. Furthermore, implementation of intraoperative magnetic resonance (MR) imaging significantly enhances the extent of resection, and, consequently, the OS [11]. In general, resection of the tumor should be maximized whenever possible—in some cases the tumor is located in eloquent areas and the volume of resection should be carefully decided. Therefore, surgical treatment should be individually tailored to every patient. Based on the volume of residual tumor there are several surgical strategies: supramaximal or supramarginal resection (SMR), gross total resection (GTR) or complete resection of contrast-enhancing tumor, near total resection (NTR), subtotal resection (STR) of the tumor, and biopsy [7] (see Table 1).

### 2.1. Supramaximal Resection

The concept of SMR is based on the fact that GBM cells are present even beyond the contrast-enhancing regions of conventional imaging techniques, such as MR or positron emission tomography (PET) imaging [7,12,13,14]. The resection of GBM is, therefore, extended to surrounding tissue beyond contrast-enhancing margins. Several studies have described better outcomes in patients who underwent GBM resection with partial removal (roughly 50%) of T2/FLAIR-hyperintense abnormal zone [7,15,16]. Even though the extent of resection in SMR is still not well defined, class III evidence points to better OS and PFS in patients undergoing this surgical strategy [17,18]. It should be emphasized that in many cases SMR is not desirable due to possible proximity of the tumor to eloquent brain areas. In order to better elucidate potential benefits of SMR compared to other neurosurgical strategies of GBM resection, more well-designed studies are needed with clear definitions of technical, radiological, and outcome measures [18].

### 2.2. Gross Total Resection

GTR has been a primary strategy for GBM resection for a long time. It is conventionally defined as complete removal of the tumor mass [7]. However, contemporary studies challenged this definition—due to pronounced invasiveness of GBM in surrounding tissue, complete removal is improbable [19]. Although usage of GTR as a resection strategy persisted, more appropriate term for GTR would be complete resection of contrast-enhancing tumor, as suggested by Karschnia et al. [7]. Similar to SMR, the minimal desired percentage of tumor removal with this neurosurgical resection strategy is not well defined. Some studies suggest that at least 90% of the contrast-enhancing tumor volume should be resected to be considered GTR [16]. However, most studies consider 100% removal of contrast-enhancing tumor as GTR [7,15,20,21,22]. Despite the assumption that differences in the 90–100% range might seem negligible at first, striking differences in outcomes are observed even for this range of resection [23]. Overall, inconsistent definition of GTR should be noted and better definition, i.e., complete resection of contrast-enhancing tumor, should be used in further studies to achieve better insights into outcomes of this neurosurgical resection strategy.

### 2.3. Near Total Resection

The concept of NTR is conventionally used when at least 80% of the GBM is removed but, at the same time, the resected volume is less than 100% of the contrast-enhancing tumor [7]. However, the range from 80 to 99% is rather large and noteworthy differences in outcomes have been described. In particular, patients with at least 95% of resection exhibited better outcomes in terms of survival and recurrence rate compared to patients who had less than 95% of the tumor removed [24,25]. It should be noted that in the study by Sanai et al. [25] outcome differences were observed even in the 95–100% range. In terms of absolute values, patients who had less than 1 cm^3^ of residual contrast-enhancing tumor on postoperative MR scan had the best outcomes [26]. Based on these findings, Karschnia et al. [7] suggested that only resection of at least 95% of contrast-enhancing tumor should be considered as NTR. Similar to the problem with GTR, inconsistent definitions of NTR limit the ability to appropriately compare the results to other surgical strategies. Nonetheless, for NTR relevant studies have reported the best outcomes for patients with at least 95% of the contrast-enhancing tumor removed or less than 1 cm^3^ of residual tumor on postoperative imaging.

### 2.4. Subtotal Resection

STR pertains to the reduction of the tumor where a substantial part of the residual is still present after the procedure [7]. There is an ongoing debate about the minimal proportion of removal which provides a clinical benefit to the patient. The most commonly suggested threshold of GBM resection for improved outcomes is roughly 80% of the contrast-enhancing tumor [7,15,23,27]. It is argued that the term STR should be used when less than 5 cm^3^ of the residual contrast-enhancing tumor is present on postoperative imaging [24,26]. This strategy is mostly utilized with large GBM which infiltrates crucial brain centers or to relieve the mass effect of the tumor itself. Any extent of resection below these thresholds is generally considered as a partial reduction of the tumor or biopsy.

### 2.5. Biopsy

Many patients with GBM have poor general health with low scores on the Karnofsky performance scale (KPS) [28]. Biopsy as a surgical strategy does not provide therapeutic benefits per se but is used as a tool for obtaining tumor samples for diagnosis and to guide further treatments. As the biopsy is less invasive procedure than resections, it is mostly utilized in older patients or other patients with low KPS who would have significant difficulties tolerating more invasive surgery [29]. Additionally, biopsy is also often utilized for patients presenting with inoperable GBM, where obtaining the tissue sample is the main goal of the surgery.

### 2.6. Technical Improvements in Surgical Treatment of Glioblastoma

As previously mentioned, numerous advances have already been achieved with the goal of improving the extent of safe resection of GBM. The implementation of neuronavigation systems during brain surgery became a common practice worldwide. This particular technology improved the extent of resection, and, concurrently, OS of patients with GBM [30,31]. Another important advancement to surgical treatment of GBM is the technique of fluorescence-guided surgery [10]. Among fluorescent agents, the most widely used and studied is 5-aminolevulinic acid (5-ALA). In randomized controlled trials, fluorescence-guided surgery with 5-ALA improved the extent of resection, as well as PFS of patients with GBM [32,33]. Another important fluorescent agent is fluorescein—observational studies suggest improved outcomes with fluorescein-guided resection as well [34]. There are many molecular targeting agents in development with encouraging results, however, as of yet these agents have not been implemented in everyday practice [34]. A novel development which might significantly improve the extent of resection is the intraoperative mass spectrometry [35]. In particular, desorption electrospray ionization (DESI) mass spectrometry could be an invaluable intraoperative tool to determine the presence of tumor cells in a studied sample, with almost real-time feedback [36]. Overall, major technological improvements have already been achieved to advance the surgical treatment of GBM, and it is encouraging that many novel advancements are still being developed.

In patients where the tumor is located in close proximity to crucial neural structures, such as the primary motor cortex or speech centers, awake craniotomy is often utilized. Combining awake craniotomy with cortical and subcortical motor mapping, image guidance, and augmented reality was already successfully used for the resection of motor cortex glioma by Turcotte et al. [37]. Interestingly, several studies utilized machine learning to better define GBM margins and infiltrative regions for improved radiotherapy regimens [38,39,40]. Theoretically, a similar approach could be used to better guide the resection of GBM. As previously mentioned, supramaximal resection is recommended as a surgical strategy for GBM when possible. However, resection beyond the contrast-enhancing region of the tumor is not well defined [7]. Correctly identifying surrounding areas of tumor infiltration via advanced machine-learning models could enhance resection and potentially improve outcomes.

## 3. Radiation Therapy (RT)

As previously discussed, due to the infiltrative nature of GBM, resection of the tumor in the vast majority of cases is not curative. Postoperative residual tumor cells contribute to GBM recurrence [41] and further treatments are necessary to decrease the probability of a relapse and improve outcomes of these patients. Radiation therapy (RT) presents an important component of GBM treatment. Current RT guidelines by the European Association of Neuro-Oncology (EANO), the Joint Tumor Section of the American Association of Neurological Surgeons (AANS), and the Congress of Neurological Surgeons (CNS) are summarized in Table 2. RT is routinely utilized as a standard adjuvant treatment after surgical resection—randomized trials have described improved outcomes in patients receiving postoperative RT compared to patients who received supportive treatment [42,43]. Importantly, for unresectable GBM in unfavorable locations RT can be used as a main treatment option [41]. Initial studies evaluating the impact of RT on outcomes of patients with GBM suffered from inconsistencies in their designs. To be more specific, in most cases patients were not randomized and, as we are now aware, radiation doses were relatively low to provide a meaningful effect [44,45]. However, a case series from 1966 by Uihlein et al. [46] suggested improved outcomes in patients who received an average total dose of radiation in the range 5000–6000 cGy. This finding ignited further studies evaluating the role of RT in GBM treatment. Finally, the Brain Tumor Study Group (BTSG) designed randomized studies investigating the impact of post-resection RT in patients with GBM which demonstrated favorable outcomes [42,43,47]. The aforementioned trials by this group played a major part in establishing post-resection RT as the standard of care for GBM patients [41], which is still used.

### 3.1. Initial Challenges of Radiation Therapy for Glioblastoma

Although the aforementioned studies described improved outcomes of patients undergoing RT, the question of the radiation dosage arose. An analysis of dose-response in RT of GBM, Walker et al. described negligible effect of radiation below 4500 cGy [50]. This, in turn, inspired further studies to investigate radiation dose escalation. In general, following studies demonstrated that doses greater than 6000 cGy did not result in significantly improved outcomes [51]. At the same time, increasing radiation dose beyond 6000 cGy resulted in greater toxicity, i.e., radiation necrosis [52].

At the time, GBM was considered as a malignancy with multiple origin sites [53,54]. In line with this assumption, a common strategy of radiation was whole-brain RT (WBRT) [41]. As the understanding of the pathophysiology of GBM advanced, the concept of multicentric involvement with GBM was abandoned—autopsy and radiographic findings suggested that the vast majority of GBM relapses occur within 2 cm of the initial location [55]. Consequently, this challenged the conventional strategy of WBRT. Therefore, the focus was shifted towards more anatomically defined RT which would be less toxic to the healthy brain. A new strategy of RT was developed, so-called involved-field RT (IFRT), which was defined as RT of the tumor itself and a provisional geometric margin of adjacent brain tissue [56,57,58,59]. Due to decreased toxicity and similar outcomes compared to WBRT, IFRT is still used as a standard of care for RT for GBM [41]. However, there are many controversies with this approach which will later be discussed.

The novel RT strategy consisted of two phases—in the first phase WBRT with 3000–4600 cGy dose was administered, after which followed a second phase of boost IFRT of lower dose (2000 to 3000 cGy) to the tumor [60,61,62]. Implementation of CT and MR imaging in RT planning permitted clearer definition of the tumor distribution, and, consequently, more precise RT [63], which, in turn, enabled more accurate administration of RT in two phases [64]. In the initial phase, radiation was delivered to the T1-contrast-enhancing tumor and adjacent edema with margins increased by 2 cm. The boost phase consisted of RT of only T1-contrast-enhancing tumor with an additional 1 cm margin. Significantly, this strategy produced similar outcomes and reduced toxicity compared to WBRT [64]. However, the concept of predetermined margins for RT, which was inspired by population-level research, presents with many flaws [38]. The margin distance from the contrast-enhancing tumor significantly differs even between official RT guidelines [65]. Furthermore, as we now know from biopsies and post-mortem studies, malignant cells in GBM do not invade adjacent brain uniformly [66,67]. In addition, the degree of invasiveness of those cells is highly specific to a patient—therefore, patients do not have the same benefits of RT with predetermined margins [38]. Despite many controversies, IFRT is still the standard RT regimen for GBM worldwide.

### 3.2. Improved Radiotherapy Regimens for Glioblastoma

In order to reduce the toxic effects of radiation to the surrounding brain tissue, intensity-modulated RT (IMRT) was developed [64]. A notable advantage of IMRT compared to the conventional RT was the ability to administer a high radiation dose to the tumor, while preserving as much healthy brain as possible. This is achieved by simultaneous delivery of multiple radiation beams from various directions to the tumor which results in different radiation dosage in the target tissue. An important benefit of IMRT is more exact tumor radiation, as well as reduced toxicity [68,69,70]. When compared to IFRT, in many cases IMRT permits reduction of the margin for RT [68]. Notably, when planning for IFRT prior imaging is performed to reduce any errors [64]. The advantage of IMRT to other regimens is most evident in GBM located close to delicate structures, such as the brainstem or the optic chiasm. In these scenarios, the radiation dose of IMRT is altered to reduce the toxic effects to neighboring structures.

Another regimen that is increasingly being studied and sometimes used for GBM is hypofractionation. The main idea behind hypofractionation is the increase in dose intensity in the target issue—this is achieved by administering fewer and larger doses of RT per treatment, which result in shorter duration of the whole RT regimen compared to other strategies [71]. However, meta-analysis by Trone et al. [71] determined that hypofractionation resulted in similar outcomes to standard regimens in different patient groups, including the elderly. The shorter duration of hypofractionation is mostly utilized for patients with poor condition or the elderly, which might have difficulties with standard regimen.

Volumetric modulated arc therapy (VMAT) is a relatively novel radiation technique with the unique ability to simultaneously alter three parameters during the process of RT, i.e., gantry speed, dose rate, and field aperture [72]. By changing these parameters, it allows for a more precise conformal dose delivery to the target tissue with significant reduction of the radiation dose to peripheral regions [73]. It has been successfully used in treatment of liver and prostate cancer [74,75]. In a study by Cheung et al. [73], VMAT regimen displayed high quality of the treatment plan, and, concurrently, significant reduction of radiation to organs at risk. Results from this particular study suggest that VMAT could be a better option than IFRT for GBM located in close proximity to orbit, hypothalamus, optic chiasm, and other delicate structures.

### 3.3. Personalized Radiotherapy for Glioblastoma

In order to reduce adverse effects of RT to normal brain tissue, it is imperative to improve the process of pretreatment planning. Implementing modern imaging techniques in the planning process, such as MR and PET scanning, and in particular diffusion tensor imaging (DTI) tractography results in a specificity of 81% and sensitivity of 98% to precisely localize the gross tumor, tumor infiltration, and healthy brain [39].

On the other hand, implementing computational tumor models to accurately estimate the tumor infiltration seemed to be a promising strategy to guide the patient-specific RT regimen. Unfortunately, the translation of this approach to everyday practice was faced with many limitations. However, machine learning might overcome these limitations. In particular, in a study by Lipkova et al. [38], the Bayesian machine-learning framework was used in order to calibrate GBM growth models from multimodal radiographic imaging—MR and PET scans. This approach enabled the generation of accurate estimates of patient-specific tumor cell density which can be used to design personalized RT regimen. According to the authors, these estimates can better guide the definition of margin for each patient but, at the same time, possible zones within GBM for dose escalation. Another avenue that is increasingly explored to better define the infiltrative region of GBM for RT planning is the utilization of deep learning. In an interesting study, Peeken et al. [40] used deep-learning-based free water correction of DTI scans to define the area of infiltration in GBM. Implementing this regimen in clinical practice could reduce margins and individualize RT for patients. Furthermore, besides improving RT planning, integrating mathematical modeling in everyday practice could serve us to better predict the response to treatment [76]. Although these systems are still in their infancy, they offer encouraging initial results. Machine-learning models could also be leveraged to predict pseudoprogression versus progression [77] Admittedly, we still lack randomized controlled trials of these frameworks and their comparison to the current standard of care. Interestingly, the addition of certain compounds might enhance the effects of RT, as well as chemotherapy. One of these compounds is resveratrol which provides inherent anti-tumoral effect, but, at the same time, improves the effectiveness of aforementioned therapies [78]. In addition, radiosensitizers, such as high-Z metal nanoparticles, could also increase the effectiveness of RT [79].

## 4. Chemotherapy

The idea of incorporating chemotherapeutic agents in the treatment of GBM stems from research led by the BTSG in 1960s. In these multi-center randomized clinical trials, BCNU (1,3-bis(2-chloroethyl)-1-nitrourea) was used as a sole agent—the administration of BCNU resulted in improved outcomes compared to the standard of care at that time [43,80]. This ignited further research in the area of chemotherapy for GBM [51]. Although chemotherapy presents a third and important pillar for GBM treatment in recent times, generally, results are modest at best. Unfortunately, promising results derived from pre-clinical trials have not been replicated in patients, and, at the moment, there is no curative chemotherapeutic treatment for GBM.

### Currently Available Drugs

At the moment, there are only three drugs which are approved by the Food and Drug Administration (FDA): temozolomide (TMZ), bevacizumab, and BCNU (also known as carmustine). It should be noted that bevacizumab is not considered a chemotherapeutic agent but will be described in the following section. Treatment regimens and outcomes of these drugs are presented in Table 3.

TMZ is the most widely used chemotherapy option for the treatment of GBM [88]. It is a DNA alkylating agent that crosses the blood–brain barrier (BBB) and is routinely used concomitant to fractionated radiation therapy and thereafter as a first-line maintenance chemotherapy [89]. In randomized controlled trials compared to radiotherapy alone, the addition of TMZ improved OS, as well as PFS [81,82], which encouraged further research and, finally, the addition of TMZ as a standard of care. It should be emphasized that the effect of TMZ is largely dependent on the timing of administration—specified schedule should always be followed, as a single dose is inadequate [90]. The current protocol and the standard of care, which is widely known as the Stupp regimen [82], consists of concomitant RT and adjuvant maintenance chemotherapy with TMZ as a scheme of 75 mg per square meter of body-surface area daily from the start until the last day of RT, which is then followed by six cycles of adjuvant TMZ of 150 to 200 mg per square meter of body-surface area for five days during each 28-day cycle. Initially, the median survival of the patients following this protocol was 14.6 months compared to 12.1 months of patients treated only with RT. In a more recent study by Lakomy et al. [91], patients following the Stupp regimen had a median OS about two months longer than the patients in an original study by Stupp et al. [82]. A major obstacle with TMZ treatment is the inherent genetic or treatment-induced resistance of GBM to TMZ, which reduces the positive effect of the drug [92]. There are many proposed mechanisms for this resistance, however, currently the only reliable predictor is the methylation state of O^6^-methylguanine DNA methyltransferase (MGMT) promoter [93]. It should be emphasized that the recent CeTeG/NOA-09 trial suggests that TMZ and lomustine (also known as CCNU) can be an alternative option for newly diagnosed MGMT methylated GBM for young patients [94]. Even though the most recent Classification of the Tumors of the CNS by WHO in 2021 combined histopathological features with molecular markers [8] and significantly improved our understanding of these neoplasms, it did not translate to a major improvement in patient stratification according to their TMZ response [89]. Based on this classification, patients with isocitrate dehydrogenase (IDH) 1/2 mutations usually have better outcomes [95], although this particular malignancy is now recognized as grade 4 astrocytoma [8].

One of the hallmarks of GBM is the process of angiogenesis due to hypoxic environment within the tumor—the tumor progression is closely linked with the formation of new blood vessels [96]. So far, a variety of angiogenic factors have been associated with this process in GBM: vascular endothelial growth factor (VEGF), platelet-derived growth factor (PDGF), basic fibroblast growth factor (bFGF), hepatocyte growth factor (HGF), and others [96]. Based on this, several drugs were developed to counteract the effects of angiogenic factors, and, consequently, the progression of GBM. Among these drugs, the most prominent is bevacizumab, which is an anti-VEGF monoclonal antibody. It improves PFS but not OS [84] and is an FDA-approved drug for recurrent GBM [88]. Interestingly, a recent study by Kaka et al. [83] on patients with newly diagnosed GBM also described longer PFS but inconsistent effect on OS. These findings warrant further research of bevacizumab utility for patients with newly diagnosed GBM.

Lastly, another alkylating drug—BCNU—has historically been used as a chemotherapeutic option for GBM—in more recent times it is generally avoided due to its toxic systemic effects [88,97]. In order to avoid the systemic adverse events, which are most evident on the bone marrow, liver, and kidney [97], the idea of local delivery of BCNU emerged. Therefore, so-called BCNU wafers implants were developed and implanted within resection cavity as a part of neurosurgical procedure [98]. This local delivery of BCNU resulted in a statistically significant increase in median survival of these patients, as presented in a meta-analysis by Chowdhary et al. [99]. The major setback of this treatment modality is the notable increase in periprocedural complications [99]. However, the idea of a local drug delivery within the resection cavity of the tumor spurred a new area of GBM research.

## 5. Novel Therapeutic Options

As previously mentioned, notable progress in the treatment of GBM has been rather slow. Despite currently the best standard of care, patients diagnosed with GBM have a median survival of just 15 months [1]. Therefore, scientific focus intensified towards novel treatment options with some encouraging results.

### 5.1. Tumor-Treating Fields (TTFields)

TTFields are alternating electrical fields of low intensity and intermediate frequency which are generated by a specialized device mounted on the shaved scalp of a patient [100]. These electrical fields have the ability to interrupt rapid cell division within the tumor, and, therefore, halt further tumor growth [101]. Significantly, TTFields only have minor side effects, mainly of dermatological nature due to the wearable nature of the device [102], as these fields do not affect healthy, non-dividing neural cells [103]. Randomized clinical trials reported longer OS and PFS in patients using TTFields with adjuvant TMZ [102,104]. Unfortunately, routine implementation of TTFields as GBM treatment is still limited by high financial cost [105].

### 5.2. Vaccine-Based Immunotherapies

Although vaccination is most often utilized in the prevention of infectious diseases, it could be a valuable tool to combat many malignancies. Vaccines work by enabling a long-term immunity against a specific antigen, which could also be a tumor-specific antigen (TSA). At the moment, there are two FDA-approved vaccines used to prevent cancer—human papilloma virus (HPV) vaccine, which is used to prevent urogenital cancers, and hepatitis B vaccine (HepB), which could prevent HepB-associated liver cancer [106]. A significant scientific focus was directed towards vaccine-based therapy for GBM.

One of the initial ideas was the development of peptide-based vaccines. It was hypothesized that specific proteins encoded by mutated genes were produced exclusively in tumor cells [107]. In theory, inducing long-term immunity against these TSAs could be used to eliminate those tumor cells. Unfortunately, proteins in GBM are not highly tumor-specific, and generating immunity against those proteins might lead to undesired autoimmune reactions [108]. Another obstacle is a significant heterogeneity of tumor cells within GBM in the same patient [35]. Therefore, progress has been slow, but advances have still been achieved. In particular, EGFT type III mutant is the most frequently observed TSA in GBM but is found in only 20–30% of GBM [107]. Although initial results were encouraging [109], more recent studies were not as promising, which slowed down further development of EGFR-based vaccines [110]. As previously mentioned, IDH mutations are specific to grade 4 astrocytoma [111] which could be leveraged to develop a vaccine. Indeed, IDH-based vaccines induce antigen-specific CD4+ T cells and humoral immunity [112]. Clinical trials of these vaccines are still under way [107]. DCs are antigen-presenting cells which can induce strong immune responses to a specific antigen [113]. Autologous DCs are derived in vitro from peripheral blood mononuclear cells (PBMC) which can then be primed by tumor-specific antigens in culture. Vaccination of patients with these primed DCs could induce antitumor immunity. Autologous DC vaccine, which is pulsed with lysate derived from GBM stem-like cell line, is safe and a well-tolerated immunotherapy option [114]. Phase II clinical trials described improved outcomes in patients receiving DC vaccines [115,116] but further studies are still ongoing. On the other hand, injection of formalin-fixed tumor cells is also evaluated. It was reported in a phase I/IIa trial that patients receiving this type of vaccine had a median PFS of 8.2 months and median OS of 22.2 months [117]. In a recent phase III trial, Liau and associates described clinically relevant and statistically significant survival extension in patients who received lysate-loaded DC vaccine and standard of care when compared to patients who only received the standard of care [118]. 

In conclusion, vaccine immunotherapies are often limited by targeting a single antigen, which might not be a viable strategy to treat GBM because of its high heterogeneity [35]. This problem could be solved by developing personalized GBM vaccines. Surgically removed GBM sample could be used for creating these vaccines, as is already evaluated for lung cancer with promising results [119]. Furthermore, a study by Hilf et al. [120] integrated highly personalized vaccines based on multiple tumor antigens which successfully elicited antitumor response. However, more well controlled studies evaluating this concept for GBM treatment are needed.

### 5.3. Oncolytic Viral Therapy

Another innovative approach to GBM treatment is the utilization of oncolytic viruses. This modality uses carefully engineered viral particles which could be able to terminate tumor cells with oncolysis, as well as induce greater immune response to the tumor itself [121]. Another therapeutic utility is using virus as a delivery vehicle for specific genes which could be implemented into tumor cells [122]. Combining oncolytic viral therapy, immunotherapy, and targeted molecular treatment could have a synergistic effect and improve outcomes [123]. The majority of oncolytic viral therapy is still in the early phases of clinical trials [100]. However, in a study by Todo et al., intratumoral administration of oncolytic herpes simplex virus type 1 resulted in improved outcomes and good safety profile [124]. It should be noted that, according to the most recent trials, the virus-based immunotherapies should be mostly considered as an adjunct treatment option, and not as a single treatment [125]. The efficacy of oncolytic viral therapy for GBM can also be improved with manipulation of specific intracellular pathways [126]. It is plausible that identifying signaling pathways involved in response to oncolytic therapy within tumor cells will also enhance the efficacy of this treatment modality. As the bioengineering technology further develops, oncolytic viral therapy could become even more effective and tumor-specific [121].

### 5.4. T-Cell Immunotherapy

A significant improvement in the area of immunotherapy for many cancers has been achieved but this is still not as effective for GBM. Patient-derived T cells can be transduced with TSA chimeric antigen receptor (CAR), and these cells are commonly known as CAR T cells [127]. Viability of CAR T cell-based therapies for GBM is faced with many obstacles, including limited permeability across the BBB, tumor microenvironment which is immunosuppressive, potential neurotoxicity, and many others [128]. Overcoming these limitations is crucial to further progress of CAR T-cell based immunotherapy for GBM. In a recent study, Wang et al. [129] described that combining oncolytic viral therapy with CAR-T cells treatment could have a synergistic effect on GBM. Modifying local immune microenvironment could enhance the efficacy of CAR-based immunotherapy [130]. Interestingly, natural killer (NK) cells, which are actively suppressed by tumor cells, can be activated to elicit antitumor response. In particular, CAR NK cells are also developed for the treatment of malignant glioma [131]. Generally, CAR NK cells are safer than CAR T cells but have a shorter life span and repeated treatments might be utilized [131].

Immune checkpoint inhibitors (CPIs) are also able to enhance the immune response to the malignancy [132] and several have been FDA-approved for various cancers [133]. Unfortunately, studies evaluating feasibility of CPIs for GBM have not yielded encouraging results so far [134].

### 5.5. Drug Repurposing

As previously stated, potential therapeutic options for GBM need to cross the BBB to reach therapeutically effective concentrations within the tumor region. Therefore, tools for drug discovery have been focused on candidates with special physicochemical characteristics which would permit crossing of the BBB [135]. Additionally, novel drug-delivery options, such as nanoparticles and focused ultrasound sonification, have been utilized to improve drug penetration within the tumor [136,137]. However, the development of novel drugs is a lengthy and costly process [138], and the idea of repurposing existing drugs for GBM has recently emerged [139]. In this context, many existing drugs have been examined in clinical trials for their anti-cancer properties in GBM treatment [140,141,142].

Among these, antipsychotic drugs have drawn a lot of interest [142]. Interestingly, anecdotal reports described improved outcomes in patients diagnosed with GBM who took antipsychotics [143,144]. Furthermore, several studies reported lower incidence of cancer among patients suffering from schizophrenia taking this class of drugs [143,145,146,147]. Antipsychotic drugs affect many intracellular signaling pathways which have a major role in GBM growth and progression [142]. In addition, these drugs interfere with many neurotransmitters and neuromodulators, as well as their receptors, within the brain which are involved in proliferation of the tumor cells [148,149]. Many other effects of these drugs on GBM cells are still under investigation [150,151,152,153]. However, antipsychotic drugs are a great example of drug repurposing for GBM—these medications are generally inexpensive, and they have relatively predictable and dose-dependent side-effect profiles [154]. Hence, clinical trials evaluating the implementation of these drugs with the current standard of care would not be as intensive process as designing new drugs from the very beginning, and could be worth pursuing [142].

Repurposing drugs which are used in melanoma treatment is also under investigation for GBM. To be more specific, the combination of dabrafenib and trametinib showed improved clinical outcomes in patients with low-grade and high-grade gliomas with specific mutation [155]. Furthermore, disulfiram which is FDA-approved drug for the treatment of chronic alcoholism might increase the sensitivity of glioma cells to chemotherapy [156]. Drug-screening platforms have suggested actinomycin-D as a promising candidate for GBM treatment [157]. Unfortunately, clinical trials have yet to investigate this particular drug most likely due to its low CNS penetration, which could be largely improved with novel drug-delivery options [139]. Ultimately, with the advent of multi-omics technologies even more already existing drugs will be evaluated for their anti-GBM properties [158].

### 5.6. Sequencing Techniques

As previously mentioned, the most recent WHO classification of CNS tumors combined histological and molecular characteristics of tumors [8]. Molecular studies of GBM significantly improved our understanding of its pathophysiology. Genomic characterization of GBM defined three intracellular pathways that were significantly altered: the p53 pathway, the Rb pathway, and the receptor tyrosine kinase (RTK)/Ras/phosphoinositide 3-kinase (PI3K) pathway [159]. Unfortunately, clinical trials targeting these pathways did not provide satisfactory results [160]. However, the interest in characterizing molecular and genomic peculiarities of GBM remained and high-throughput array technologies, such as next-generation sequencing (NGS), whole-exome sequencing (WES), and whole-genome sequencing (WGS), have been utilized in studying GBM [161]. This resulted in classification of GBM based on genomic and transcriptomic data, and the three distinct subtypes have been proposed, namely, proneural, mesenchymal, and classical [159]. This particular classification has been central in the process of associating molecular attributes to clinical phenotypes [161]. Modern sequencing technologies usually analyzed only one tissue sample from a single GBM, which was inadequate due to inherent heterogeneity of GBM, as was evident by RNA-sequencing of the tumor [162]. Consequently, improved genomic approaches have to be used when studying GBM in order to characterize the whole tumor, and not just a specific subpopulation within the tumor. Sequencing technologies are perfect candidates for this task which could provide us with new molecular biomarkers and potential therapeutic targets [161]. Studies using these sequencing techniques identified various intracellular pathways and biomarkers, but also genomic differences between treated and untreated tumors [163]. Similarly, epigenomic analysis identified that promoter-methylation of the DNA-repair gene MGMT was associated with better response to TMZ treatment [93]. Furthermore, epigenomic analysis identified biomarkers which were predictive of tumor recurrence and prognosis [164]. Overall, high-throughput techniques, which are mentioned above, are likely to play a crucial role in the future of personalized treatment of GBM. They have the potential to better predict treatment response and improve our development, and, later, the selection of specific molecular-targeted therapies.

## 6. Concluding Remarks

Although biomedicine has improved significantly over the years in treating many conditions, including cancers, the tangible progress in GBM treatment is still lacking. Currently, the standard of care consists of maximal safe neurosurgical resection, followed by radiochemotherapy and chemotherapy. When considering individualized therapy for GBM, we often think of novel treatment options, however, as presented in this review, even surgery, radiotherapy, and chemotherapy can be personalized to an extent. In particular, with our better understanding of GBM pathophysiology and more precise classification, in the near future chemotherapeutic agents could be personalized based on molecular and metabolic characteristics of the tumor [165]. Indeed, precision medicine approaches are likely to play a crucial role in the treatment of GBM. These mutation-based profiling tools could be utilized to identify tumor-specific actionable mutations and drive personalized therapies [166]. Liquid biopsy could significantly contribute to this approach—it would provide clinicians a safe and less invasive way of gaining regular insights into tumor adaptations and potential treatment options [167]. It is speculated that extracellular vesicles, due to their diverse information related to the tumor could be the key to precision medicine of GBM [168]. Furthermore, another interesting innovation are patient-derived tumor organoids which could be used for tailored chemotherapeutic approach and to predict the tumor response to specific agents [169]. Compared to conventional GBM models, these organoids preserve inherent heterogeneity and resistance to treatment [170] and could be utilized to develop new compounds and evaluate their effectiveness. Interestingly, these patient-specific organoids could be evaluated based on their sensitivity to already existing drugs, such as the ones we described earlier which could improve outcomes. It is plausible that artificial intelligence and machine learning could play a major role in suggesting personalized treatment options based on multi-omics data derived from the tumor samples. Another promising avenue is the development of NGS technologies which could effectively identify genetic, molecular, and epigenetic signatures of the tumor and drive precision medicine and highly individualized therapies [171]. NGS is able to identify genetic mutations within the studied sample tissue which could affect clinical decision-making [172]. Despite the great potential of NGS technologies, it is still generally underutilized in clinical practice and decision-making with GBM patients [173], and, even when NGS is used, at this point it still does not significantly affect the treatment trajectory [172]. Even though many advancements in our understanding of GBM pathophysiology have already been achieved, the standard of care remains unchanged. Current standard of care and potential future therapeutic options are presented in Figure 1.

In conclusion, GBM is still treated insufficiently, and the treatment is not yet personalized. Certain novel therapeutic options provide promising results, but these must be evaluated in well-designed studies before they can be implemented in everyday practice.

## Figures and Tables

**Figure 1 biomedicines-11-01579-f001:**
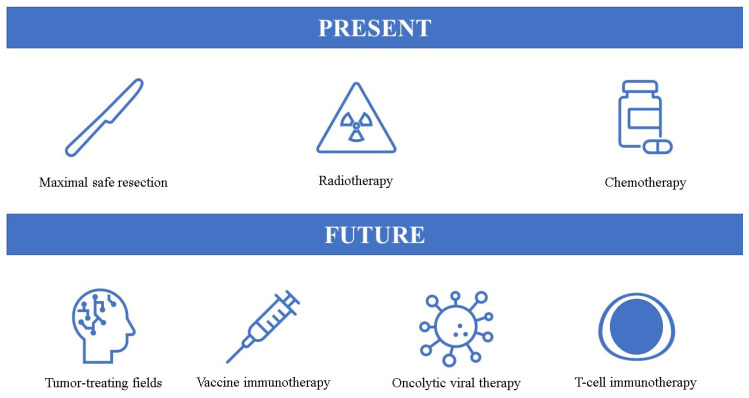
Present and potential future options for glioblastoma treatment.

**Table 1 biomedicines-11-01579-t001:** Neurosurgical strategies for glioblastoma resection.

Neurosurgical Strategy	Extent of Resection
Supramaximal resection	Complete resection of contrast-enhancing tumor and partial resection of T2/FLAIR-hyperintense abnormal zone
Gross total resection	Complete resection of contrast-enhancing tumor
Near total resection	Resection of ≥95% of contrast-enhancing tumor
Subtotal resection	Resection of <95% of contrast-enhancing tumor
Biopsy	Diagnostic procedure without resection

**Table 2 biomedicines-11-01579-t002:** Current guidelines on the radiotherapy of glioblastoma.

**EANO Guidelines on the Radiotherapy of Glioblastoma [48]**	
Timing:	RT should start within 3–5 weeks after surgery
Dosing:	1.8–2 Gy daily fractions for a total of 50–60 Gy
	Hypofractionated RT is recommended for elderly and patients with KPS < 70
Margin:	1–2 cm beyond the tumor area identified by MRI sequences
	Another smaller margin (0.3–0.5 cm) is added to compensate for uncertainties
	Highly sensitive structures should be outlined
Follow-up:	MRI scan 3–4 weeks after completion of RT
**AANS/CNS guidelines on the radiotherapy of glioblastoma [49]**	
Timing:	RT should start within 6 weeks after surgery
Dosing:	2 Gy daily fractions with the standard dose of 60 Gy
	Hypofractionated or hyperfractionated RT is recommended for frail and elderly patients *
Margin:	1–2 cm beyond the tumor area identified by MRI sequences
	Recalculation of the radiation volume during RT is suggested
Follow-up:	No specific recommendation

AANS/CNS, American Association of Neurological Surgeons/Congress of Neurological Surgeons; EANO, European Association of Neuro-Oncology; Gy, gray; KPS, Karnofsky performance scale; MRI, magnetic resonance imaging; RT, radiotherapy. * Schemes of 40.05 Gy administered in 15 fractions or 25 Gy in 5 fractions or 34 Gy in 10 fractions should be considered.

**Table 3 biomedicines-11-01579-t003:** Dosage regimens and reported outcomes of FDA-approved drugs for glioblastoma.

Drug	Regimen	Outcomes
Temozolomide	75 mg/m^2^ daily concurrent with RT 150–200 mg/m^2^ for 5 days during each 28-day cycle	Improves OS and PFS for newly diagnosed GBM [81,82]
Bevacizumab	10 mg/kg every 2 weeks	Improves PFS for recurrent and newly diagnosed GBM [83] Inconsistent effect on OS [84]
BCNU	150–200 mg/m^2^ daily every 6 weeks	Improved OS after relapse [85] Inconsistent effect on OS and PFS [86]
BCNU implant	8 × 7.7 mg intracranially implanted wafers for a total of 61.6 mg	Improved OS but not PFS of newly diagnosed GBM [87]

GBM, glioblastoma; OS, overall survival; PFS, progression-free survival; RT, radiotherapy.

## Data Availability

Not applicable.

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
