# Peer review of "Personalized Treatment of Glioblastoma: Current State and Future Perspective"

_biomedicines, 2023, doi:10.3390/biomedicines11061579_

Round 1
Reviewer 1 Report
The goal of this manuscript is to establish a catalog of the current standard of care as well as novel personalized treatments for GBM.
the manuscript is well written and the cited references are appropriate. The review is useful in the field for an update of the current treatment and personalized therapies in perspective. The subject is well discussed. However the title of the review mentions "future perspective". In this particular point, the discussion has not gone far enough. Indeed, more about the possible causes of the therapeutic failure should be discussed as well as possible pertinent combined therapies that could be used to improve the therapeutic efficiency, even if it is not yet tested in clinical trials. The author should add a paragraph dedicated to this point and make this review even more interesting to the field.
Author Response
Dear Reviewer, thank You for Your comments.
We have expanded on future perspectives in lines 157-167, 286-293, 404-423.
Combined therapies have been added in lines 288-292, 335-338, 412-415.
Kind regards, Authors.
Reviewer 2 Report
This review paper aimes to highlight personalized approaches in GBM treatment.
The main focus is the description of the current therapeutic approaches for GBM treatment., whereas the "future perspectives" are rather short.
In the first paragraph they summarize surgical strategies such has complete , supramaximal or partial resections. While this is done in much detail, it is not clear how these approaches are used in a personalized manner. According to the title of the paper this should be the main focus, currently it is not.
In the paragraph on biopsy it is not mentiones that also tumors that are inoperable are indications for biopsies, not only poor patient status.
The general conclusion that "many novel (surgical) advancements" are being developed, is very unspecific, exactly this should be explained in detail.
In the radiotherapy section, too much emphasis is put on the history of this treatment (again without any reference to personalization) and the most interesting part -personalized radiotherapy- is too short.
In the chemothetrapy section, corrections should be made. It is unclear why BCNU is in the focus but CCNU which often serves as the drug in the control arm of multiple trials is not mentioned at all. Furthermore and with a personalized approach, in patients with methylated MGMT promotor the combination of TMZ and CCNU can be considered (CETEG-trial), this needs to be included. Furthermore, Bevacizumab is not chemotherapy and should not be mentioned as such.
When the authors review DC-based approaches and virotherapy, they should include the latest results of two phase-III trials in the respective fields by Liau and Todo, the statement that the field has not progressed beyond phase-II is incorrect.
The authors should also mention the option to individualize GBM treatment based on next-generation sequencing and the potential drugs that could be used based on the molecular findings.
Author Response
Dear Reviewer,
Thank You for Your comments.
We have expanded on future perspectives and the possibility to individualize tumor resection in lines 157-167.
We have included the suggested indication for biopsy for inoperable tumors in lines 132-134.
As previously mentioned, surgical novelties are now further discussed in lines 157-167.
We have expanded on individualized RT in lines 286-293.
Thank You for this correction. We have made the suggested adjustments regarding CCNU and CeTeG trial in lines 335-338.
We have also corrected our error of classifying bevacizumab as chemotherapy, please see lines 307-308.
The latest results from studies by Liau and Todo have also been added in lines 412-415 and 428-340. Thank You for the references.
Finally, we have added lines 465-468 regarding next generation sequencing.
Kind regards,
Authors.
Reviewer 3 Report
The review article by Rončević et al. is a well-written, comprehensive overview of current therapy protocols and future treatment strategies in glioblastoma. I have only minor comments
Throughout: please, remove space between number and %
-Introduction, first line: Glioblastoma (GBM) is the most common primary malignancy of the brain in adults,..
-Introduction, 6th line: ….line treatment, which consists of surgery, concomitant radiochemotherapy, and maintenance chemotherapy
-Page 2, paragraph 2 and Page 3, last paragraph: please introduce interoperative MRT
-Page 6, last sentence: I would not classify bevacizumab as a chemotherapeutics
-Page 7, paragraph 2: temozolomide concomitant to fractionated radiation therapy and thereafter as first line maintenance chemotherapy
-Page 7, second paragraph, line 9: synchronized RT????? Better: concomitant to radiotherapy and adjuvant maintenance chemotherapy
-Page 7, end of second paragraph and page 8, middle of last paragraph: according to the current WHO classification, IDH-mutated tumors are now classified as grade 4 astrocytoma and not any more as glioblastoma
-Page 8, first paragraph: Please, add that temozolomide-lomustine combination therapy is an alternative treatment option in newly diagnosed MGMT methylated glioblastoma for young patients in, e.g., Germany (J Neurooncol. 2023 Jan;161(1):147-153. doi: 10.1007/s11060-022-04203-4.)
-Page 8, last paragraph: please, describe in more detail autologous dendritic cell-based immunotherapy (expansion of autologous DC cells from patient-derived PBMCs, stimulations with lysates of patient-derived stem cell-enriched glioblastoma cultures)
-Same paragraph: Please, give also some information about the current developments in personalized peptide vaccination strategies (e.g., Nature. 2019 Jan;565(7738):240-245. doi: 10.1038/s41586-018-0810-y. )
Page 9: Please, also mention CAR-NK cell therapy approaches in glioblastoma (e.g., Front Immunol
. 2019 Nov 14;10:2683. doi: 10.3389/fimmu.2019.02683. eCollection 2019.)
Concluding remarks: .... followed by radio-CHEMOtherapy and chemotherapy.
Author Response
Dear Reviewer,
Thank You for Your comments.
We have removed space between number and % throughout the manuscript.
We have also added "in adults" in line 38.
We have corrected the suggested lines in the introduction regarding treatment, please see lines 43-44.
Intraoperative MRI is now introduced in lines 65-66.
We made a mistake of classifying bevacizumab as a chemotherapeutic. Please see our correction in lines 307-308.
The correction "temozolomide concomitant to fractionated radiation therapy and thereafter as first line maintenance chemotherapy" is now included in lines 318-319.
"Synchronized RT" has now been excluded and Your suggestion was added, lines 324-325.
Thank You for the correction based on the most recent classification of CNS tumors. The correction is made in line 343.
We have added temozolomide-lomustine combination therapy as an alternative treatment option in lines 335-338.
We have expanded on DC immunotherapies at length in lines 404-415.
We also provided some extra information and included the suggested reference in lines 420-421 and 428-430.
CAR NK cells are now described in lines 443-446.
Suggested correction in concluding remarks has been added in line 456.
Kind regards,
Authors.